# A two-tiered latent class and spatial analytical approach to identify clusters of neonatal mortality among very low birth weight infants: A population-based cohort study

Daniela Testoni Costa-Nobre[1]*, Adriana Sanudo[1], Kelsy Nema Areco[1], Ana Sílvia Scavacini Marinonio[1], Milton Harumi Miyoshi[1‡], Tulio Konstantyner[1‡], Carina Nunes Vieira e Oliveira[1‡], Rita de Cassia Xavier Balda[1‡], Mandira Daripa Kawakami[1‡], Paulo Bandiera-Paiva[1‡], Rosa Maria Viera Freitas[2‡], Mônica La Porte Teixeira[2‡], Bernadette Cunha Waldvogel[2‡], Maria Fernanda de Almeida[1], Ruth Guinsburg[1], Carlos Roberto Veiga Kiffer[1,3]

1 Escola Paulista de Medicina – Universidade Federal de São Paulo (UNIFESP), São Paulo, São Paulo, Brazil, 2 Fundação Sistema Estadual de Análise de Dados (SEADE Foundation), São Paulo, São Paulo, Brazil, 3 CEPID – ARIES (Antimicrobial Resistance Institute of São Paulo), São Paulo, São Paulo, Brazil

☯ These authors contributed equally to this work.
‡ These authors also contributed equally to this work.
* danielatestoni@gmail.com

## Abstract

### Objective

To identify and analyze patterns of neonatal deaths among very low birth weight (VLBW) infants in the most socioeconomically developed state of Brazil, from 2004 to 2020, using a two-tiered probabilistic approach that combines Latent Class Analysis (LCA) and spatial analysis.

### Study design

This historical population-based cohort study included 137,224 live births with birthweight of 400-1499g to mothers residing in São Paulo State, using linked birth and death certificate data.

### Results

Among 42,230 neonatal deaths, five distinct latent classes were identified: infection-dominant, intrapartum event-dominant, malformation-dominant, respiratory-dominant, and other. Survival analysis showed differences in timing of death across classes, with intrapartum-related deaths concentrated in the first hours of life, and infection-related deaths occurring later. Spatial analysis revealed geographic clustering especially for infection, malformation, and respiratory-related deaths, primarily in southern municipalities of the State.

which permits unrestricted use, distribution, and reproduction in any medium, provided the original author and source are credited.

**Data availability statement:** Data available at: https://zenodo.org/records/12696458.

**Funding:** Funding was provided by Fundação de Amparo à Pesquisa do Estado de São Paulo (FAPESP), Project #2017/03748-7. Database use was possible due to agreements #23089.004297/2008-11 and #23089.000057/2014-95 between Fundação SEADE and Universidade Federal de São Paulo. The funding agency did not interfere with or participate in the study design, data analysis, interpretation of the results, or the writing and revision of the manuscript. The authors received no other specific funding for this work.

**Competing interests:** The authors have declared that no competing interests exist.

**Abbreviations:** VLBW, very low birth weight (VLBW); LCA, Latent Class Analysis (LCA); Fundação SEADE, State Data Analysis System; ICD, International Classification of Diseases; LC, latent classes; AIC, Akaike Information Criteria; BIC, Bayesian Information Criteria; SSABIC, sample-size-adjusted BIC, VMLR, Voung-Lo–Mendell–Rubin test; LMRadjLRT, bootstrap likelihood ratio test; LISA, Local Indicators of Spatial Association.

## Conclusion

The combined use of LCA and spatial analysis identified distinct patterns of neonatal mortality. LCA differentiated clinically meaningful profiles with specific timing of death, while spatial analysis revealed municipal-level clustering and overlap of these patterns. These findings showed how neonatal mortality is shaped by both diagnostic profiles and territorial context, providing actionable evidence to guide targeted improvements in perinatal and neonatal care to reduce preventable deaths among VLBW infants.

## Introduction

Understanding the multifocal nature of neonatal mortality requires analytical approaches that can account for diagnostic uncertainty, overlapping clinical conditions, and geographic disparities in access and quality of maternal neonatal care. Despite overall improvements in health indicators in Brazil, the reduction in neonatal mortality has been slower and uneven across regions, particularly in areas with limited access to high-quality health services [1]. Neonatal mortality accounts for approximately 70% of all infant deaths in the country, reflecting persistent inequalities in perinatal care [2].

Secondary data from vital statistics systems offer valuable insights into the complex interplay of biological, health systems, and socio-environmental determinants of neonatal deaths. When appropriately analyzed, these data enable large-scale assessments of mortality patterns, the burden of specific conditions and spatio-temporal disparities [3]. However, traditional approaches often struggle to disentangle these overlapping factors and to account for the variability and uncertainty in diagnostic reporting. To address these limitations, probabilistic classification methods such as Latent Class Analysis (LCA) can classify deaths into mutually exclusive groups based on co-occurring diagnostic information, refining interpretation of overlapping causes [4]. Each major category of neonatal death has distinct etiological, clinical, and programmatic implications [5]. LCA accommodates this complexity by grouping cases according to shared diagnostic profiles, rather than relying solely on underlying cause-of-death coding.

Moreover, integrating LCA with spatial analysis and geoprocessing can identify geographic areas with elevated probabilities of clustering for specific latent classes (LC) of death, revealing spatial heterogeneity and potential hotspots of systemic failure [6]. Stratifying deaths by both diagnostic patterns and geographic location thus enhances understanding of health system performance and facilitates the design of targeted interventions. In addition, different causes of death are often associated with distinct timing of death. Intrapartum-related deaths, for instance, may reflect deficiencies in labor and delivery care, while deaths from infections may indicate failures in neonatal care or infection control [7]. Early neonatal deaths (0–6 days) are often linked to birth asphyxia or extreme prematurity, whereas late neonatal deaths (7–27 days) are more frequently associated with infections, congenital anomalies, and postnatal complications [7–9].

Therefore, this study aimed to identify and analyze latent patterns of neonatal deaths among very low birth weight (VLBW) infants in the state of São Paulo, Brazil, over the period from 2004 to 2020, applying a two-tiered probabilistic approach that combines LCA and spatial analysis.

## Methods

### Population and case definition

This was a historical population-based cohort study using data from deterministic linkage between death and live birth certificates. The study included all live births ≥22 weeks gestational age and birth weight between 400 and 1499 grams, born from 2004 to 2020 to mothers residing in São Paulo State, Brazil. A deterministic linkage process was applied to link live births and neonatal deaths (<28 days). [10] Data were received from the State Data Analysis System Foundation (Fundação SEADE) on 17/01/2024, which compiles vital statistics with over 99% coverage of births and deaths in São Paulo [11]. We had full access to the de-identified population-based database used to create the study population. Procedures to ensure data adequacy, including data preparation and data validation prior to data use, were undertaken and have been previously published [12].

Diagnostic groups were defined a priori to represent major etiological pathways of neonatal mortality and were intentionally broad to capture overlapping diagnoses, allowing latent class analysis to probabilistically identify clinically meaningful patterns of co-occurring conditions [5,7–9]. Diagnoses from the death certificates were categorized using the 10th revision of International Classification of Diseases (ICD-10) [13] codes into the following groups (full list in S1 Table):

- Infection: bacterial and fungal sepsis, meningitis, pneumonia (A32.7; A40; A41; B37.7; G00-01; G04; J13-18; M86; N30; N39.0; P23; P36; P37.2; P38-39).

- Intrapartum events: birth injury, intrauterine hypoxia, birth asphyxia and meconium aspiration syndrome (P10-15; P20; P21; P24.0).

- Malformations: any malformation listed in the death certificate (Q00-99).

- Respiratory failure or diseases: respiratory conditions of the perinatal period, pneumothorax, respiratory failure, extreme immaturity, respiratory distress syndrome of newborn, interstitial lung diseases (J43; J81; J84; J93; J96; P07.2; P22; P25; P27-28).

Multiple diagnoses could be assigned to a single live birth. The study was approved by the Ethics Committee of the Federal University of São Paulo (#65016822.1.0000.5505). The requirement for informed consent was waived by the Institutional Review Board due to the use of secondary data and a de-identified database.

### First-tier: Latent class analysis

The diagnostic groups served as the basis for the first-tier approach of LCA. LCA was applied to identify distinct subgroups within the cohort of neonatal deaths in the entire population [14]. Models ranging from one to six LCs were estimated, using 600 random starting values and 120 final stage optimizations to ensure model stability and avoid local maxima. Model selection was guided by multiple fit indices, including the Akaike Information Criteria (AIC), Bayesian Information Criteria (BIC), sample-size-adjusted BIC (SSABIC), the Voung-Lo–Mendell–Rubin test (VMLR), the bootstrap likelihood ratio test (LMRadjLRT), and entropy [15].

Among the information criteria AIC, BIC, and SSABIC indicated better model fit; in cases where the criteria suggested different solutions, preference was given to the model with the lowest BIC, in accordance with previous recommendations [15]. The model with the highest entropy was also favored, as entropy provides a summary measure of classification precision, with values closer to one indicating greater certainty in class assignment and separation between LCs. When

the VLMR and bootstrap LMRadjLRT tests yield divergent results, the latter was prioritized due to its superior accuracy in model selection [15]. While all statistical criteria were considered, the final model choice also incorporated expert judgment, prioritizing solutions that were conceptually meaningful with internally consistent and distinct LCs.

Once the number of LCs had been calculated and the final model chosen, the probability of each neonate's class was estimated, and they were classified into the most appropriate LC. Finally, for clarity each LC was numbered and named based on the predominant diagnosis of its specific class.

### Second-tier: Spatial analysis

The neonatal mortality rate for each LC was calculated at the municipal level, based on the municipality of death, by dividing the total number of neonatal deaths in each class by the total number of live births, expressed as per cent of live births. These rates were calculated for each municipality and a smoothing method was applied with a local Bayesian estimator for reducing the impact of outlier areas with low event counts by considering the mean rates of its neighbors [16,17]. Spatial dependency and autocorrelation of the neonatal mortality rates for each LC per municipality were assessed using Global Moran's Indicator (*I*) and Local Indicators of Spatial Association (LISA). Finally, a visual correlation was performed between neonatal mortality clusters for each pair of LCs to identify overlapping patterns across municipalities. Different colors were used to represent distinct cluster combinations (e.g., both LCs showing "high-high" or "low-low", or one LC showing "high-high" and the other "low-low"). This visual representation enabled the identification of spatial overlaps across municipalities.

### Validation of early and late mortality categorization

The Kaplan–Meier survival analysis was conducted as a sensibility test to estimate and compare the timing of death (in days of life) among neonates assigned to each LC identified in the final model. This approach enabled visualization and statistical comparison of survival curves, providing objective evidence to confirm whether specific LC were predominantly associated with early neonatal deaths (0–6 days) or late neonatal deaths (7–27 days).

### Data management

Data management was performed using STATA software (version 18.5, StataCorp LP, College Station, TX, USA). LCA analysis was conducted using the Mplus program (version 8.11) [18] and spatial analysis was conducted with TerraView software (version 4.2.2, Instituto Nacional de Pesquisas Espaciais, São José dos Campos, Brazil). Geographic coordinates and spatial files for São Paulo State and its municipalities were obtained from the Brazilian Institute of Geography and Statistics [19].

## Results

### Population

From 2004 to 2020, a total of 10,265,105 live births were recorded. Of these, 9,713,206 had a birth weight of 1500g or more, 410,221 had an unknown birth weight, 2341 had birth weight < 400g, and 2113 had a gestational age < 22 weeks, resulting in 137,224 newborns included in the study. Among them, 42,230 (31%) died during the neonatal period. Maternal age was under 20 years in 17%, 60% were born by cesarean section and 34% were born before 28 weeks of gestation.

### First-tier: Latent class analysis

A total of seven LC models (1–7 classes) were evaluated (Table 1). As expected, the 1-class solution showed the poorest fit to the data. Although the 7-class model yielded the lowest AIC, BIC, and SSABIC values, its classes were not conceptually distinct or interpretable. The entropy of the 7-class model (0.879) was only marginally higher than that of the 5-class

**Table 1. Comparison of model fit statistics for selection of optimal latent class model.**

|  | AIC | BIC | SSABIC | p-VLMR-LRT | p-LMRadjLRT | Entropy |
|---|---|---|---|---|---|---|
| **1 class** | 1,059,243.628 | 1,059,321.486 | 1,059,292.884 | – | – | – |
| **2 classes** | 1,039,349.218 | 1,039,496.283 | 1,039,442.256 | <0.001 | <0.001 | 1.000 |
| **3 classes** | 1,030,334.696 | 1,030,550.968 | 1,030,471.518 | <0.001 | <0.001 | 0.866 |
| **4 classes** | 1,023,466.872 | 1,023,752.351 | 1,023,647.477 | <0.001 | <0.001 | 0.859 |
| **5 classes** | 1,018,010.573 | 1,018,365.218 | 1,018,234.919 | <0.001 | <0.001 | 0.875 |
| **6 classes** | 1,017,540.970 | 1,017,964.863 | 1,017,809.141 | <0.001 | <0.001 | 0.845 |
| **7 classes** | 1,015,351.691 | 1,015,844.792 | 1,015,663.645 | <0.001 | <0.001 | 0.879 |

AIC: Akaike Information Criteria; BIC: Bayesian Information Criteria; SSABIC: Sample-size-adjusted BIC; VMLR: Voung-Lo-Mendell-Rubin test; LMRad-jLRT: Bootstrap likelihood ratio test.

model (0.875), and comparable to that of the 6-class model (0.845). Despite the slightly lower SSABIC observed in the 6-class model, the 5-class model also demonstrated adequate fit indices.

The 5-class model was chosen, and each LC was named based on the predominant diagnosis: class 1 as infection-dominant, class 2 as intrapartum event-dominant, class 3 as malformation-dominant, class 4 as respiratory-dominant, and class 5 as other. Regarding the distribution of the population among LCs, 28% (n = 11,758) belonged to LC 1; 4% (n = 1834) to class 2; 9% (n = 3643) to class 3; 46% (n = 19,601) to class 4 and 13% (n = 5394) to class 5. LC 1 comprised exclusively of infants with an infection-related diagnosis in any line of the death certificate, with 8% also having intrapartum events, 8% malformations, and 27% respiratory issues. LC 2 had no infection-related diagnoses, but all deaths were associated with intrapartum events, with 1% also presenting malformations and 3% respiratory issues. LC 3 included 7% of infants with an infection-related diagnosis, 15% with intrapartum events, 100% with malformations, and 32% with respiratory issues. In LC 4 all cases had respiratory issues, and 23% of infants also with an infection-related diagnosis, 14% with intrapartum events, and 2% with malformations. LC 5 had infants only with other causes of death, with no diagnoses related to infection, intrapartum events, malformations, or respiratory issues (Fig 1). Demographic characteristics for each class are described in Table 2.

## Second-tier: Spatial analysis

The spatial analysis focused on the four most significant LC, i.e.,: infection-dominant (1), intrapartum event-dominant (2), malformation-dominant (3) and respiratory-dominant (4). Class 5 (other) comprised diverse infrequent and/or not discriminative diagnoses and was not used in the second-tier analysis. Descriptive maps displaying neonatal mortality rates by quintiles for each LC suggested the presence of clustered areas across all classes (Fig 2). Moran's Global Index, applied to the smoothed rates for each class, confirmed significant spatial autocorrelation: $I = 0.50$ for infection-dominant LC ($p = 0.001$), $I = 0.30$ for intrapartum event-dominant LC ($p = 0.001$), $I = 0.43$ for malformation-dominant LC ($p = 0.001$) and $I = 0.44$ for respiratory-dominant LC ($p = 0.001$).

The LISA Cluster Maps highlighted areas of high and low rates for each LC (Fig 3). Infection, malformation and respiratory-dominant classes exhibited an apparently similar clustering pattern, which differed from that of intrapartum event-dominant LC. High-rate clusters for infection, malformation, and respiratory-dominant classes were predominantly found in the southern region of the state, whereas low-rate clusters for intrapartum event-dominant class were concentrated in the southern and central areas. In the northern and northwestern regions, alternating areas of high- and low-rate clusters were observed for each LC.

Fig 4 confirms a correlation between infection and malformation-dominant cluster areas, with a positive correlation in 40 municipalities and a negative correlation in only 4. The findings also reinforce the occurrence of a negative correlation between intrapartum event and infection, malformation and respiratory-dominant classes clusters.

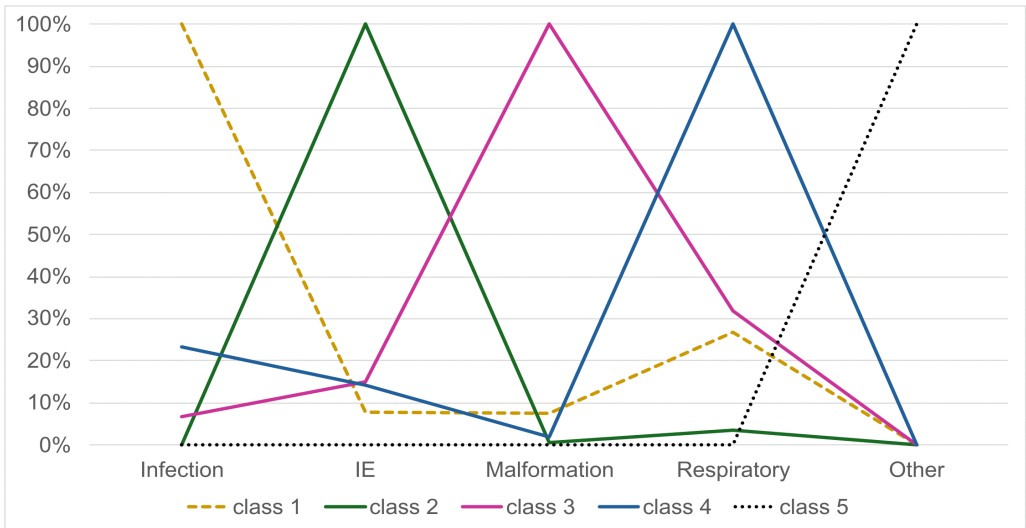

**Fig 1. Overlay plot of latent classes categorized by causes of death.** IE: intrapartum event.

**Table 2. Demographic characteristics across five latent classes of death and survival.**

| | Infection-dominant | Intrapartum Event-dominant | Malformation-dominant | Respiratory-dominant | Other | Survival |
|---|---|---|---|---|---|---|
| Maternal age* | | | | | | |
| <20 years | 20% | 21% | 15% | 21% | 21% | 15% |
| 20–34 years | 64% | 64% | 63% | 65% | 64% | 66% |
| ≥35 years | 16% | 15% | 22% | 14% | 14% | 19% |
| Pre-natal visits+ | | | | | | |
| Absent | 7% | 10% | 3% | 8% | 10% | 4% |
| 1–3 | 23% | 39% | 17% | 30% | 31% | 17% |
| 4–6 | 47% | 42% | 39% | 46% | 42% | 44% |
| ≥7 | 23% | 17% | 41% | 16% | 17% | 35% |
| Multiple gestation* | 15% | 15% | 10% | 17% | 17% | 18% |
| Cesarean section* | 53% | 38% | 61% | 41% | 38% | 66% |
| Gestational age+ | | | | | | |
| <28 weeks | 42% | 61% | 23% | 78% | 69% | 21% |
| 28 to <32 weeks | 44% | 30% | 38% | 18% | 23% | 48% |
| ≥32 weeks | 14% | 9% | 39% | 4% | 8% | 31% |
| Male* | 55% | 55% | 51% | 55% | 55% | 49% |

*>99% valid information, +87% valid information.

## Validation of early and late mortality categorization

As shown by Fig 5, time of death varied across LCs. Deaths occurred significantly later in the infection-dominant class, with a median of 8 days or 190 hours (interquartile range, IQR: 77–339 hours), while they occurred earlier in the intrapartum event-dominant class, with a median of 7 hours (IQR: 1–48 hours). The median time to death was 11 hours (IQR: 1–96 hours) for the malformation-dominant class and 37 hours (IQR: 8–96 hours) for the respiratory-dominant class.

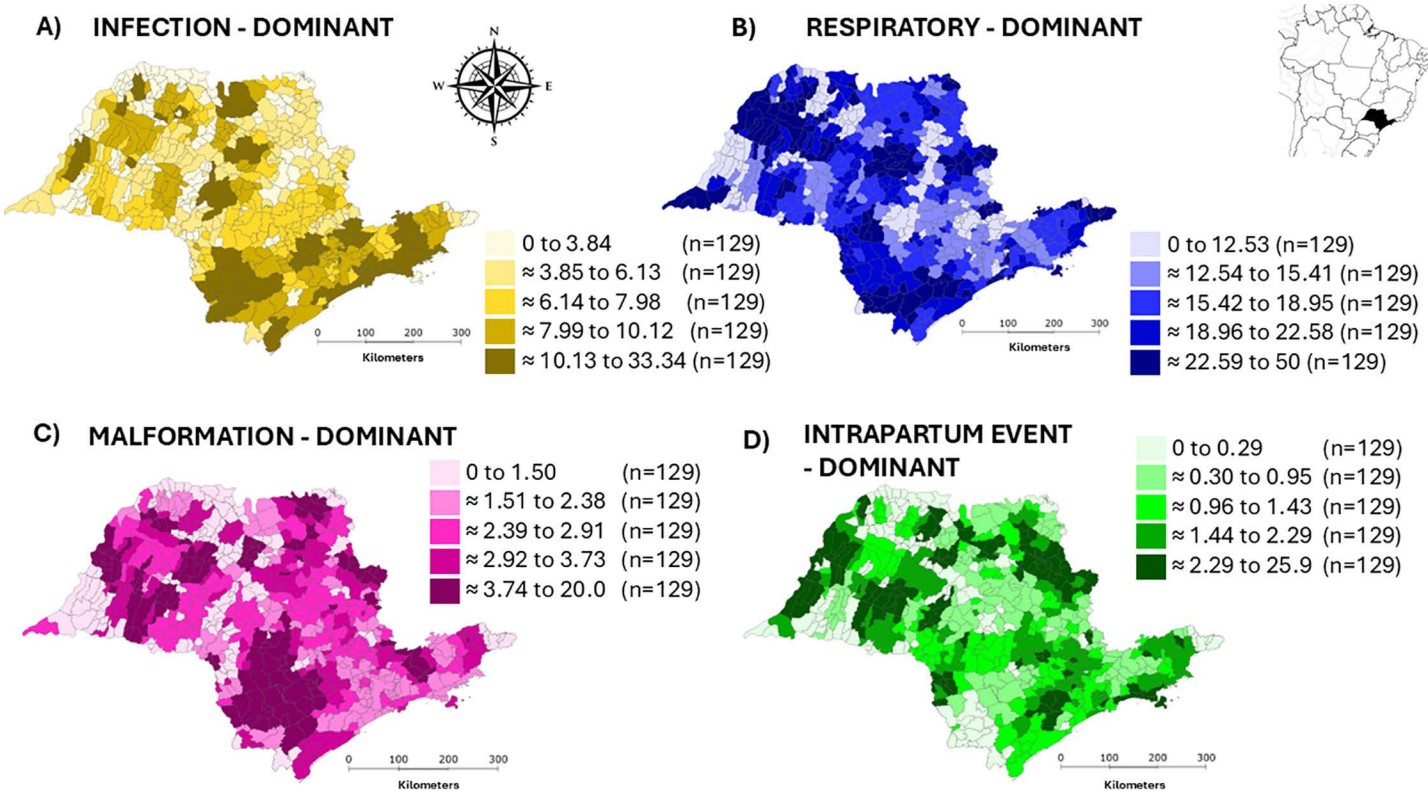

**Fig 2. Distribution of neonatal mortality rates by latent classes after smoothing with local Bayes estimates (Quintiles, %).** Latent classes: **A)** Infection-dominant, **B)** Respiratory-dominant, **C)** Malformation-dominant, and **D)** Intrapartum Event-dominant. Data from 645 São Paulo Municipalities (Brazil), from 2004-2020.

## Discussion

This study applied a two-tiered probabilistic framework, combining LCA and spatial analysis, to investigate diagnostic and geographic patterns of neonatal mortality among VLBW infants in São Paulo State, Brazil. Five distinct LCs of neonatal death were identified based on diagnostic profiles – each reflecting co-occurring clinical conditions rather than single-cause attributions – enhancing interpretability beyond conventional death certificate categorizations. Notably, one class (intrapartum-dominant) was clearly linked to early neonatal mortality, while the infection-dominant class was associated with later deaths. The malformation- and respiratory-dominant classes presented an intermediate pattern, with median time to death under 48 hours, but survival distributions that more closely resembled the infection class. Spatial clustering of mortality by class revealed significant geographic heterogeneity, particularly for infection-, malformation-, and respiratory-related deaths, which concentrated in southern municipalities of São Paulo state. These findings underscore the interaction between cause-specific mortality patterns and regional disparities in health system performance.

From a methodological perspective, the use of LCA enabled the classification of neonatal deaths into internally consistent and conceptually distinct LCs. Similar approaches have been employed in adult and children mortality studies in other settings, demonstrating LCA's utility in uncovering hidden diagnostic patterns and clarifying heterogeneous mortality profiles [20,21]. These classes reflected co-occurring diagnostic profiles and allowed for more nuanced interpretation beyond the limitations of single-cause attribution from death certificates [14,22]. The advantage of LCA is that it defines subgroups by considering multiple variables concurrently, independent of the outcome [14].

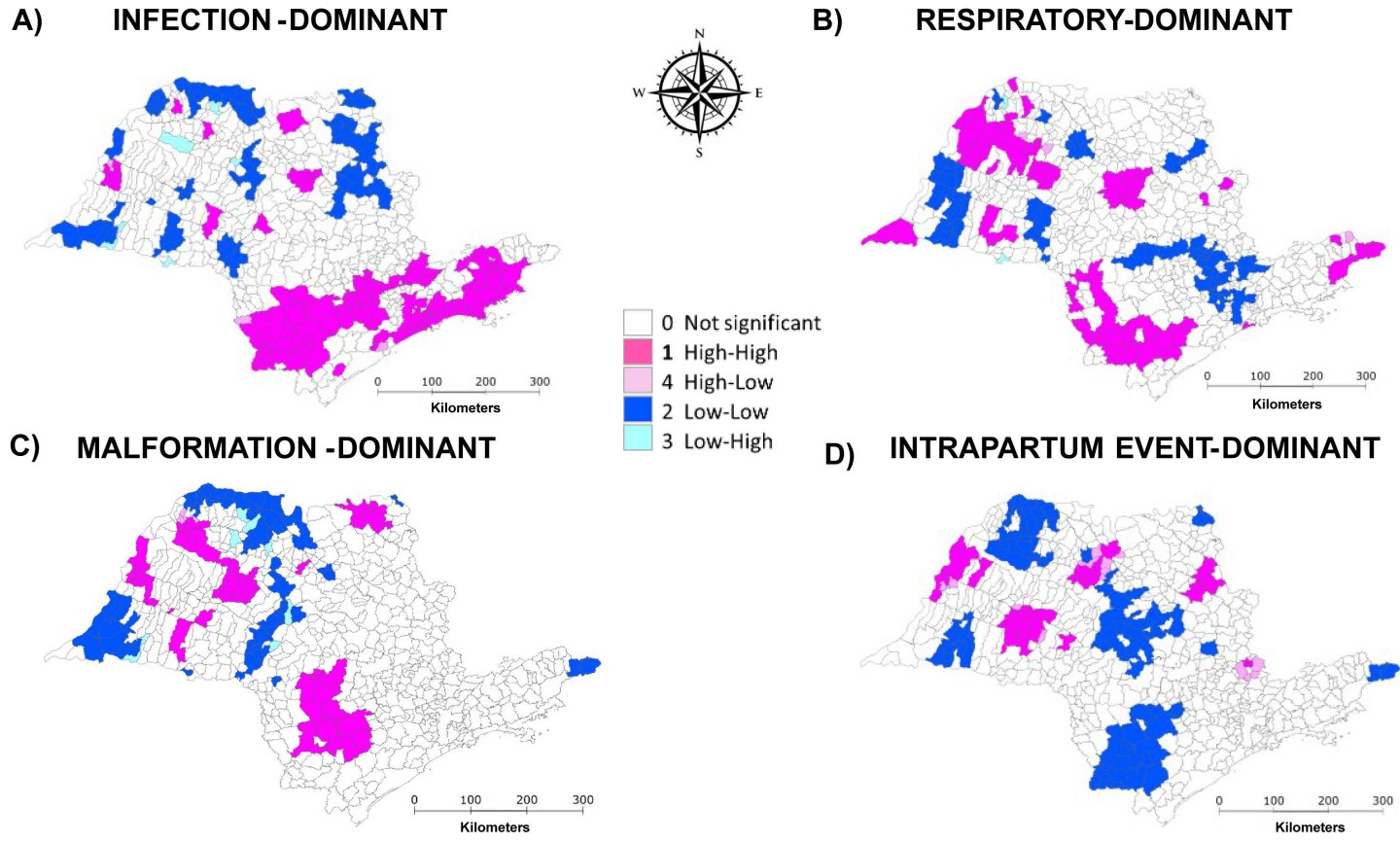

**Fig 3. LISA cluster map of neonatal mortality rates by latent classes in São Paulo State, Brazil (2004–2020).** Latent classes: **A)** Infection-dominant, **B)** Respiratory-dominant, **C)** Malformation-dominant, and **D)** Intrapartum Event-dominant. Data from 645 São Paulo Municipalities (Brazil), from 2004-2020.

The use of entropy and fit indices (AIC, BIC, SSABIC) provided statistical rigor, while the incorporation of conceptual coherence ensured the interpretability of the resulting classes [15]. The selection of the 5-class model was guided by the principle of parsimony and, most importantly, by epidemiological interpretation of the resulting classes in the context of neonatal mortality. While the 6- and 7-class solutions offered marginal statistical improvements, they lacked meaningful differentiation between classes. The 5-class model offered the best balance between statistical adequacy and substantive interpretability, making it the most appropriate choice for this analysis. The final five-class model included distinct profiles: infection-dominant, intrapartum event-dominant, malformation-dominant, respiratory-dominant, and other causes.

The second analytical tier revealed spatial heterogeneity in the distribution of these LCs across municipalities. The use of Moran's I and LISA statistics, commonly applied in spatial epidemiology, strengthened the detection of local clustering and regional disparities in mortality burden [16,23,24]. The empirical Bayesian method was used to calculate the smoothed rates, to stabilize and reduce the high variability observed in the crude rates [25]. The spatial autocorrelation findings confirmed the presence of geographic clustering, particularly in the southern region of the state for infection, malformation, and respiratory-dominant classes. In contrast, intrapartum-related deaths showed a distinct pattern, with low-rate clusters concentrated in the south-central areas. These spatial insights suggest that the burden and likely contributing factors for each class vary geographically, underscoring the need for regionally tailored public health strategies [23,25].

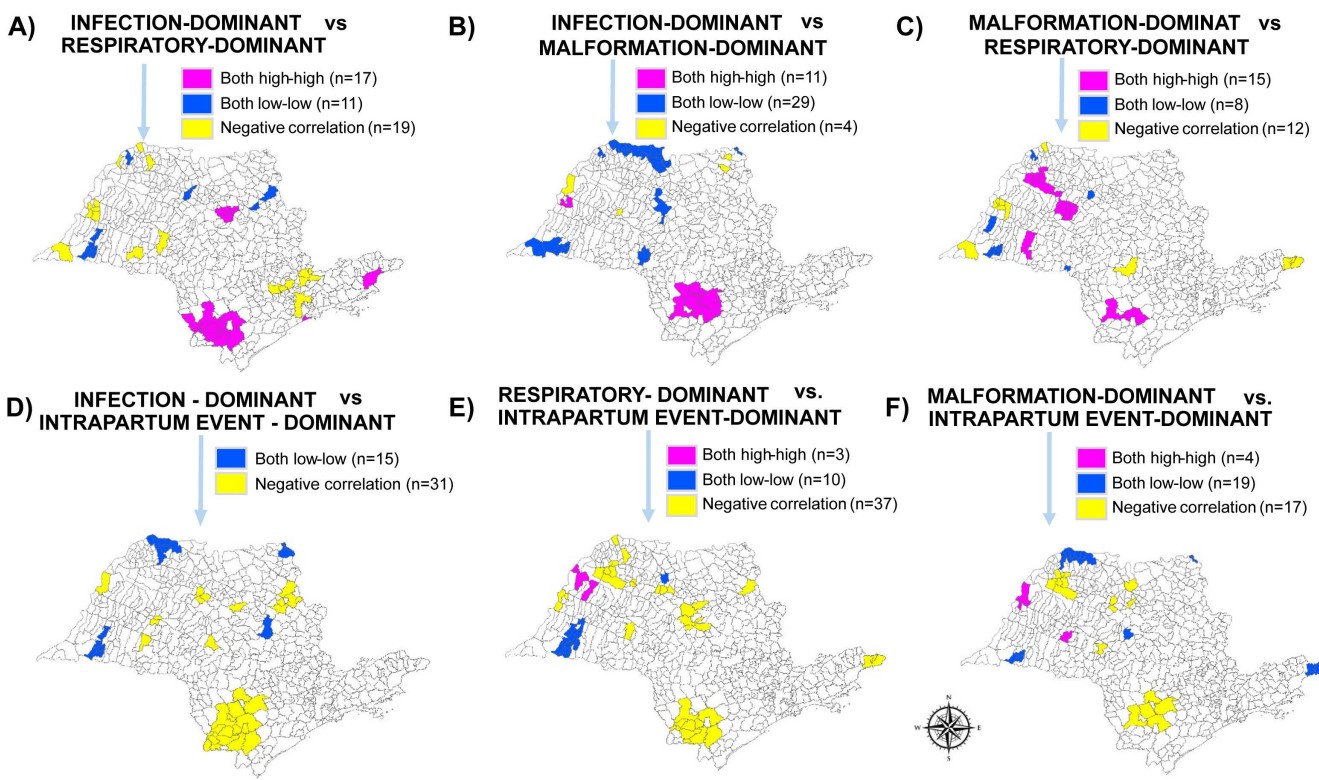

**Fig 4. Correlation of municipalities with overlapping latent class clusters of neonatal mortality in São Paulo State, Brazil (2004–2020).** Correlation among latent classes: **A)** Infection-dominant & Respiratory-dominant, **B)** Infection-dominant & Malformation-dominant, **C)** Malformation-dominant & Respiratory-dominant, **D)** Infection-dominant & Intrapartum Event-dominant, **E)** Respiratory-dominant & Intrapartum Event-dominant, and **F)** Malformation-dominant & Intrapartum Event-dominant.

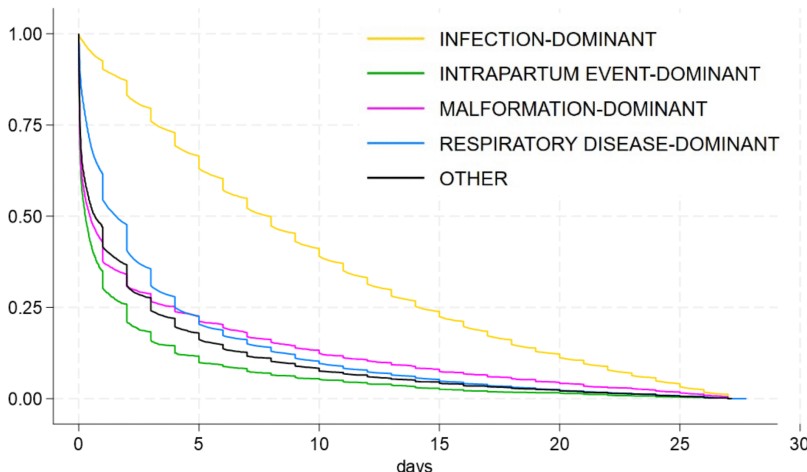

**Fig 5. Survival curve by latent class of neonatal mortality in São Paulo State, Brazil (2004–2020).**

The spatial analysis indicated that latent classes dominated by intrapartum-related conditions and infections were the main contributors to geographic disparities in neonatal mortality, as they showed the most consistent and concentrated high–high clusters across municipalities. These patterns likely reflect deficiencies in the quality of antenatal and intrapartum care, delays in referral, and limitations in neonatal intensive care and infection prevention practices [23–25]. In contrast, classes characterized predominantly by congenital malformations and respiratory conditions exhibited more diffuse or overlapping spatial patterns, suggesting a lesser contribution to overall spatial inequality. By linking class-specific spatial clustering to plausible health system gaps, these findings provide actionable guidance for targeted interventions aimed at reducing preventable neonatal deaths.

The spatial analysis may be influenced by referral bias, as clustering based on municipality of death can overestimate rates in municipalities hosting referral centers, particularly for infection- and respiratory-dominant classes associated with later neonatal deaths. Conversely, early intrapartum-related deaths may be underestimated in these areas, as deaths often occur before inter-municipality transfer. Importantly, class-specific spatial clustering offers actionable insights: intrapartum-dominant clusters point to the need for strengthening antenatal and intrapartum care, whereas infection-dominant clusters highlight priorities for infection prevention bundles, antimicrobial stewardship, and enhanced surveillance within neonatal intensive care units.

A conceptual strength of the study lies in its dual approach to distinguishing early and late neonatal deaths – drawing from both epidemiological literature and empirical survival analysis. Early neonatal deaths (0–6 days) are typically associated with perinatal complications such as extreme prematurity, birth asphyxia, and delivery-related trauma, while late neonatal deaths (7–27 days) more frequently reflect infections or congenital anomalies [5,7–9,26]. The Kaplan-Meier analysis broadly aligned with these expectations but also revealed important nuances across classes. Specifically, the intrapartum-dominant class showed a sharply concentrated distribution of deaths in the first hours of life. In contrast, the infection-dominant class exhibited a more delayed pattern, consistent with late neonatal mortality and the known pathophysiology of hospital-acquired or postnatal infections [27]. Interestingly, the malformation-dominant and respiratory-dominant classes demonstrated a median time to death of less than 48 hours, yet with distributions less accurately skewed toward the first hours than the intrapartum class. These profiles may be best characterized as occupying a transitional space between early and late neonatal deaths. Despite the earlier median timing of the malformation and respiratory dominant classes, their overall survival curves more closely resembled that of the infection class than the intrapartum class – suggesting that, although pathophysiologically distinct, these deaths may share causal pathways such as insufficient access to early specialized care, limited postnatal stabilization care, lack of timely surgical approach for congenital anomalies or inadequate ventilatory support for respiratory failure in VLBW infants [28–30]. These findings reinforce the critical role of immediate newborn care, special newborn care with dedicated staff, specialized equipment in preventing avoidable clinical deterioration among high-risk neonates [29].

In contrast, the late timing of infection-related deaths suggests these infants likely survived the initial high-risk perinatal period, only to succumb to complications arising during hospitalization. This finding highlights the importance of ongoing high-quality care within neonatal intensive care units [27,29]. Factors such as breaches in infection control practices, delays in recognizing clinical deterioration, and inappropriate empirical antimicrobial regimens may contribute significantly to these outcomes [29]. Ensuring optimal sepsis prevention, surveillance protocols and timely therapeutic interventions within neonatal intensive care units is essential in reducing late neonatal mortality [29].

These findings have important implications for neonatal care policies. The predominance of late deaths in the infection-dominant class highlights the need for improved postnatal surveillance, early diagnosis, and infection control practices, particularly in municipalities with identified clusters. Conversely, the concentration of intrapartum-related deaths in the early period underscores gaps in obstetric and delivery care that require urgent attention [5,26]. The geographic clustering of specific mortality profiles highlights the need for spatially target interventions, especially in the context of Brazil's decentralized health system, where health service delivery and resource allocation vary widely across municipalities [31].

Prior studies in Brazil have documented the impact of structural health inequalities, such as inconsistent access to neonatal intensive care and maternal care networks, in shaping neonatal outcomes [1]. This study also demonstrates the value of applying advanced analytical techniques to routinely collected data. This spatially informed approach is essential for pinpointing areas of concern, enabling more strategic intervention efforts and resource distribution, thus strengthening the study's implications for public policy. By leveraging probabilistic modeling and spatial epidemiology, it is possible to extract meaningful patterns that are otherwise obscured in aggregated mortality statistics. The results support a more refined understanding of neonatal mortality, offering a framework for evidence-based, regionally responsive strategies to reduce preventable deaths in vulnerable populations such as VLBW infants.

Nonetheless, the study has limitations. The use of death certificate data introduces potential for misclassification, though this was partially mitigated by incorporating the causes found in any line of the death certificate. Moreover, the spatial analysis was based on the municipality of death rather than residence, which may limit the ability to fully capture upstream social or systemic factors. This approach, however, was chosen intentionally to approximate the quality and effectiveness of perinatal care, under the assumption that, in most cases, death occurs near the (or at the same) place of birth. Nonetheless, we acknowledge that inter-municipality transfers may occur, potentially introducing some misclassification in the spatial attribution of care-related outcomes. As this study relied on routinely collected secondary data not originally designed to address the specific research question, potential misclassification, unmeasured confounding, and temporal changes in care practices should be considered when interpreting the findings. Future studies may benefit from integrating additional data sources, such as maternal and neonatal health records, to further contextualize mortality patterns.

In conclusion, by applying a two-tiered approach this study highlights how the combination of methodological rigor and epidemiological insight can uncover diagnostic and geographic patterns in neonatal mortality. LC analysis identified distinct and clinically meaningful diagnostic patterns, differentiating deaths predominantly related to intrapartum events from those associated with infections, respiratory conditions and congenital malformations, with clear differences in timing of death. Spatial analysis further demonstrated that these classes were unevenly distributed across municipalities, revealing geographic clustering and overlap of specific mortality patterns that reflect heterogeneity in health system performance. Interventions must address both the immediate quality of perinatal care, particularly for conditions associated with rapid deterioration, and the continuity of care in neonatal intensive care units, especially to prevent deaths from nosocomial infections and complications emerging after the early neonatal period.

## Supporting information

**S1 Table. ICD-10-coded diagnoses from death certificates and their frequency in the study population.**
(DOCX)

## Author contributions

**Conceptualization:** Daniela Testoni Costa-Nobre, Maria Fernanda de Almeida, Ruth Guinsburg, Carlos Roberto Veiga Kiffer.

**Data curation:** Kelsy Nema Areco, Milton Harumi Miyoshi, Paulo Bandiera-Paiva, Rosa Maria Viera Freitas, Mônica La Porte Teixeira, Bernadette Cunha Waldvogel.

**Formal analysis:** Daniela Testoni Costa-Nobre, Adriana Sanudo, Ana Sílvia Scavacini Marinonio, Maria Fernanda de Almeida, Ruth Guinsburg, Carlos Roberto Veiga Kiffer.

**Funding acquisition:** Ruth Guinsburg.

**Investigation:** Kelsy Nema Areco, Ana Sílvia Scavacini Marinonio, Milton Harumi Miyoshi, Tulio Konstantyner, Carina Nunes Vieira e Oliveira, Rita de Cassia Xavier Balda, Mandira Daripa Kawakami.

**Methodology:** Adriana Sanudo, Ruth Guinsburg, Carlos Roberto Veiga Kiffer.

**Supervision:** Carlos Roberto Veiga Kiffer.

**Validation:** Adriana Sanudo.

**Visualization:** Maria Fernanda de Almeida, Ruth Guinsburg.

**Writing – original draft:** Daniela Testoni Costa-Nobre, Adriana Sanudo, Maria Fernanda de Almeida, Carlos Roberto Veiga Kiffer.

**Writing – review & editing:** Kelsy Nema Areco, Ana Sílvia Scavacini Marinonio, Milton Harumi Miyoshi, Tulio Konstantyner, Carina Nunes Vieira e Oliveira, Rita de Cassia Xavier Balda, Mandira Daripa Kawakami, Paulo Bandiera-Paiva, Rosa Maria Viera Freitas, Mônica La Porte Teixeira, Bernadette Cunha Waldvogel, Ruth Guinsburg.

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
