## [Decision Letter · Decision Letter 0]

10 Dec 2025

Dear Dr. Costa-Nobre,

We look forward to receiving your revised manuscript.

Kind regards,

Ricardo Q. Gurgel, PhD

Academic Editor

PLOS One

Journal Requirements:

[The author(s) received no specific funding for this work.].

4. Thank you for stating the following in your manuscript:

[Fundação de Amparo à Pesquisa de São Paulo - FAPESP, Project # 2017/03748-7. The funding agency did not interfere or participate in the design of the study, data analysis, interpretation of the results, writing or revising of the manuscript.]

[The author(s) received no specific funding for this work.]

[No authors have competing interests].

Reviewers' comments:

Reviewer's Responses to Questions

**Comments to the Author**

1. Is the manuscript technically sound, and do the data support the conclusions?

Reviewer #1: Yes

Reviewer #2: Yes

Reviewer #3: Yes

2. Has the statistical analysis been performed appropriately and rigorously?

Reviewer #1: Yes

Reviewer #2: Yes

Reviewer #3: Yes

3. Have the authors made all data underlying the findings in their manuscript fully available?

Reviewer #1: Yes

Reviewer #2: Yes

Reviewer #3: Yes

4. Is the manuscript presented in an intelligible fashion and written in standard English?

Reviewer #1: Yes

Reviewer #2: Yes

Reviewer #3: Yes

Reviewer #1: The submitted study aims to identify and analyze patterns of neonatal deaths among very low birth weight newborns in the state of São Paulo, Brazil. It is a robust investigation that examined more than 100,000 live births and over 40,000 neonatal deaths over a 16-year period. Relevant patterns were identified using a concise methodology and a rigorous statistical analysis, employing two statistical approaches that are appropriate to the research topic and objectives. As expected within the Brazilian context, regional disparities and persistent problems were observed, which require attention and targeted efforts toward addressing preventable factors, as highlighted by the authors. This is a topic of utmost importance that should be brought to academic discussion.

Reviewer #2: commend the authors on a well-written and thoughtfully structured manuscript. The study employs a sophisticated methodology and presents findings with significant utility for public health and the field of perinatology.

During my review, I identified several areas with potential for enhancement, which I outline below.

1. Adherence to the STROBE Statement

To ensure comprehensive reporting and identification of best practices, I recommend the authors explicitly confirm that all items of the STROBE checklist have been addressed.

Item 4 (Title and Abstract): The manuscript's title should clearly state the study design. While the use of Latent Class Analysis implies an observational design, the specific type (which I understand to be a retrospective cohort study) should be explicitly stated in the title, or if not possible, the abstract. This is a key requirement for immediate reader comprehension.

Other Items: Certain other STROBE criteria appear to be only partially met. These include a more detailed description of the sampling method and a clearer explanation of how potential biases were mitigated in the methodology.

2. Formulation of the Conclusion

The conclusion, both in the abstract and the main text, should provide a direct and concise answer to the study's primary objective. While the results section presents the data, the conclusion must synthesize these findings to explicitly state how they address the aim of analyzing mortality patterns using the two methodologies. A summary of the distinct patterns identified for each methodology should be clearly articulated in the concluding remarks.

3. Application of the RECORD Checklist

Given that this study utilizes large, routinely collected health databases, the application of the RECORD checklist (Benchmark et al., 2015) is highly recommended. This checklist is an extension of STROBE specifically for such datasets.

* Key items to address include providing a flow diagram illustrating the patient selection and data linkage process (which could be included as a supplement).

* Furthermore, a detailed description of the data cleaning and preparation procedures is essential. This should cover the handling of missing data, inconsistencies, and coding errors, including the percentage of missing data and the methods used for its treatment (e.g., imputation, exclusion).

4. Application of the RECORD-PE Checklist

Finally, due to the perinatal focus of the research, I strongly suggest applying the RECORD-PE checklist (Perinatal Extension, Hermanussen et al., 2021). This extension is tailored for observational studies involving maternal, fetal, and newborn data.

Reviewer #3: PLOS ONE

Review

This study is well designed and methodologically robust, employing probabilistic Latent Class Analysis (LCA) to uncover unobserved subgroups of neonatal deaths that share similar combinations of characteristics. This approach adds considerable depth beyond the deterministic ICD-10 classification by revealing latent diagnostic patterns underlying the recorded causes of death. I included some suggestions for better clarification and strengthening of the paper.

Title: I would suggest adjusting the phrasing as: “A two-tiered latent class and spatial analytical approach to identify clusters of early and late neonatal mortality among very low birth weight infants”.

Abstract

Objective:

Line 27: …patterns of neonatal deaths among very low birth weight… maintain consistency between title (early/late mortality) and the abstract (general neonatal mortality)

Introduction

Line 58 spatial-temporal- better to replace with: spatio-temporal

Line 152: 34% were <28 weeks... refine as: “… 34% were born before 28 weeks of gestation.” To clarify the reference to gestational age.

Methods

The authors may need more detailed justification why the four diagnostic groups were selected (infection, intrapartum event, malformation, respiratory) as they are broad and include overlapping ICD codes.

Results

The paper would benefit from clearly identifying which classes are the main drivers of overall spatial disparities in neonatal mortality across the state.

The results are reported by class, but the implications for how these patterns translate into health system gaps remain somewhat implicit. Strengthening the link between (1) each class’s spatial clustering, (2) the likely underlying service-delivery challenges, and (3) the most relevant targeted interventions would enhance the translational value and clarity of the findings.

Discussion

The Discussion could better detail how referral bias may over- or under-estimate clustering.

Linking between each latent class and actionable strategies could be articulated more clearly. Providing class-specific examples (e.g., infection prevention bundles vs. improvements in delivery-room care) would strengthen policy relevance.

Conclusion

Given the paper’s title, the conclusion should more explicitly highlight what the study reveals about early vs late neonatal deaths across latent classes, based on the survival curves.

**Do you want your identity to be public for this peer review?** For information about this choice, including consent withdrawal, please see our Privacy Policy

Reviewer #1: No

Reviewer #2: **Yes:** MARCOS ALVES PAVIONE

Reviewer #3: No

---

## [Author Response · Author response to Decision Letter 1]

27 Jan 2026

Dear Dr. Gurgel,

Dear Academic Editor and Reviewers,

We would like to thank you and the reviewers for the careful evaluation of our manuscript contributing to the overall quality of our work. We are grateful for the positive assessment of the scientific merit, methodological rigor, and relevance of the study.

Below, we provide a detailed, point-by-point response to all comments raised by the Academic Editor and the reviewers. All changes have been incorporated into the revised manuscript and are highlighted in the tracked-changes version. Page and line numbers refer to the revised manuscript.

Responses to Journal and Editorial Requirements

Response:

We have revised the manuscript to fully comply with PLOS ONE formatting and style guidelines. File naming, structure, headings, tables, and figures were checked and adjusted according to the official PLOS ONE templates.

Discrepancies were identified between the Funding Information, Financial Disclosure, and text within the manuscript.

Response:

We appreciate the editor’s careful review of the funding information. We have taken the following actions:

• All funding-related text has been removed from the manuscript body and Acknowledgments section, in accordance with PLOS ONE policy.

• The Funding Statement has been corrected and standardized.

• The Financial Disclosure has been clarified to explicitly state the role of the funders.

[The author(s) received no specific funding for this work.].

We have taken the following actions:

• All funding-related text has been removed from the manuscript body and Acknowledgments section, in accordance with PLOS ONE policy.

• The Funding Statement has been corrected and standardized.

• The Financial Disclosure has been clarified to explicitly state the role of the funders.

Funding was provided by Fundação de Amparo à Pesquisa do Estado de São Paulo (FAPESP), Project #2017/03748-7. The funding agency did not interfere with or participate in the study design, data analysis, interpretation of the results, or the writing and revision of the manuscript. The authors received no other specific funding for this work. Use of the data was possible due to agreements between Fundação SEADE and Universidade Federal de São Paulo (#23089.004297/2008-11; #23089.000057/2014-95; #23089.120254/2019-34). The authors have declared that no competing interests exist.

4. Thank you for stating the following in your manuscript:

[Fundação de Amparo à Pesquisa de São Paulo - FAPESP, Project # 2017/03748-7. The funding agency did not interfere or participate in the design of the study, data analysis, interpretation of the results, writing or revising of the manuscript.]

[The author(s) received no specific funding for this work.]

We have adjusted as mentioned above:

Funding was provided by Fundação de Amparo à Pesquisa do Estado de São Paulo (FAPESP), Project #2017/03748-7. The funding agency did not interfere with or participate in the study design, data analysis, interpretation of the results, or the writing and revision of the manuscript. The authors received no other specific funding for this work. Use of the data was possible due to agreements between Fundação SEADE and Universidade Federal de São Paulo (#23089.004297/2008-11; #23089.000057/2014-95; #23089.120254/2019-34). The authors have declared that no competing interests exist.

[No authors have competing interests].

We have completed the Competing Interests section in the online submission form as requested and included the information in the cover letter. The statement has been updated to read as follows:

Funding was provided by Fundação de Amparo à Pesquisa do Estado de São Paulo (FAPESP), Project #2017/03748-7. The funding agency did not interfere with or participate in the study design, data analysis, interpretation of the results, or the writing and revision of the manuscript. The authors received no other specific funding for this work. Use of the data was possible due to agreements between Fundação SEADE and Universidade Federal de São Paulo (#23089.004297/2008-11; #23089.000057/2014-95; #23089.120254/2019-34). The authors have declared that no competing interests exist.

We reviewed the reviewer comments: no specific previously published works were recommended for citation.

We carefully reviewed the reference list to ensure completeness and accuracy. No retracted articles were cited in the manuscript. In response to the RECORD checklist requirements, two additional references were included to support methodological and reporting aspects addressed in the revision. The inclusions are below:

Ref 10

Waldvogel BC, Morais LCC, Perdigão ML, Teixeira MLP, Freitas RMV, Aranha VJ. Experiência da Fundação Seade com a aplicação da metodologia de vinculação determinística de bases de dados. Ensaio & Conjuntura. 2019;1:1-25

Ref 12

Areco KN, Konstantyner T, Bandiera-Paiva P, Balda RCX, Costa-Nobre DT, Sanudo A, et al. Operational Challenges in the Use of Structured Secondary Data for Health Research. Front Public Health. 2021 Jun 15;9:642163. doi: 10.3389/fpubh.2021.642163.

Reviewer #1

The submitted study aims to identify and analyze patterns of neonatal deaths among very low birth weight newborns in the state of São Paulo, Brazil. It is a robust investigation that examined more than 100,000 live births and over 40,000 neonatal deaths over a 16-year period. Relevant patterns were identified using a concise methodology and a rigorous statistical analysis, employing two statistical approaches that are appropriate to the research topic and objectives. As expected within the Brazilian context, regional disparities and persistent problems were observed, which require attention and targeted efforts toward addressing preventable factors, as highlighted by the authors. This is a topic of utmost importance that should be brought to academic discussion.

Response:

We thank Reviewer #1 for the positive and thoughtful evaluation of our study. We appreciate the recognition of its methodological rigor, analytical approach, and public health relevance, particularly regarding neonatal mortality patterns and regional disparities.

Reviewer #2

1. Adherence to the STROBE Statement. To ensure comprehensive reporting and identification of best practices, I recommend the authors explicitly confirm that all items of the STROBE checklist have been addressed.

Item 4 (Title and Abstract): The manuscript's title should clearly state the study design. While the use of Latent Class Analysis implies an observational design, the specific type (which I understand to be a retrospective cohort study) should be explicitly stated in the title, or if not possible, the abstract. This is a key requirement for immediate reader comprehension.

Other Items: Certain other STROBE criteria appear to be only partially met. These include a more detailed description of the sampling method and a clearer explanation of how potential biases were mitigated in the methodology.

Response:

To clearly identify the study design, and to include the suggestion of reviewer #3, we have revised the manuscript title from “Latent class and spatial analyses in a tiered approach for identifying cluster areas of early or late neonatal mortalities” to “A two-tiered class and spatial analytical approach to identify clusters of neonatal mortality among very low birth weight infants: a population-based cohort study.”

In addition, we explicitly stated the study design in the abstract (Page 2, Line 58) by adding the terms retrospective and cohort (“This retrospective population-based cohort study…”), and in the Methods section (Page 4, Line 120) (“This was a retrospective population-based cohort study using data from deterministic linkage…”).

We have reviewed the manuscript using the STROBE checklist to ensure comprehensive reporting of all recommended items. A completed STROBE checklist is now provided as supplementary material.

2. Formulation of the Conclusion

The conclusion, both in the abstract and the main text, should provide a direct and concise answer to the study's primary objective. While the results section presents the data, the conclusion must synthesize these findings to explicitly state how they address the aim of analyzing mortality patterns using the two methodologies. A summary of the distinct patterns identified for each methodology should be clearly articulated in the concluding remarks.

Response:

We thank the reviewer for this valuable suggestion and agree that the conclusion should more directly and concisely address the study’s primary objective. In response, we rewrote the conclusions in both the abstract and the main text to explicitly synthesize the findings and clarify how the combined methodologies address the aim of analyzing neonatal mortality patterns.

In the abstract (Page 2, Line 65), the conclusion was revised to state:

“The combined use of LCA and spatial analysis identified distinct patterns of neonatal mortality. LCA differentiated clinically meaningful profiles with specific timing of death, while spatial analysis revealed municipal-level clustering and overlap of these patterns. These findings showed how neonatal mortality is shaped by both diagnostic profiles and territorial context, providing actionable evidence to guide targeted improvements in perinatal and neonatal care to reduce preventable deaths among VLBW infants.”

In the main text (Page 16, Line 424), the conclusion was rewritten as follows:

“In conclusion, by applying a two-tiered approach, this study highlights how the combination of methodological rigor and epidemiological insight can uncover diagnostic and geographic patterns in neonatal mortality. Latent class analysis identified distinct and clinically meaningful diagnostic patterns, differentiating deaths predominantly related to intrapartum events from those associated with infections, respiratory conditions, and congenital malformations, with clear differences in timing of death. Spatial analysis further demonstrated that these classes were unevenly distributed across municipalities, revealing geographic clustering and overlap of specific mortality patterns that reflect heterogeneity in health system performance.”

We believe these revisions directly respond to the reviewer’s concern by clearly summarizing the distinct contributions of each methodology and explicitly linking the findings to the study’s primary objective.

3. Application of the RECORD Checklist

Given that this study utilizes large, routinely collected health databases, the application of the RECORD checklist (Benchmark et al., 2015) is highly recommended. This checklist is an extension of STROBE specifically for such datasets.

* Key items to address include providing a flow diagram illustrating the patient selection and data linkage process (which could be included as a supplement).

* Furthermore, a detailed description of the data cleaning and preparation procedures is essential. This should cover the handling of missing data, inconsistencies, and coding errors, including the percentage of missing data and the methods used for its treatment (e.g., imputation, exclusion).

Response:

We have addressed this recommendation in full:

The RECORD checklist has been completed and included as Supplementary Material.

We change the manuscript to better adhere to the RECORD checklist

*RECORD 6.2: Any validation studies of the codes or algorithms used to select the population should be referenced. If validation was conducted for this study and not published elsewhere, detailed methods and results should be provided.

Data preparation and validation were conducted through a complex and time-consuming process that has been previously published. For clarity, we have now added the sentence below along with the corresponding reference. (Ref 12)

We added in Page 4 Line 126: “Procedures to ensure data adequacy, including data preparation and data validation prior to data use, were undertaken and have been previously published.”

Areco KN, Konstantyner T, Bandiera-Paiva P, Balda RCX, Costa-Nobre DT, Sanudo A, et al. Operational Challenges in the Use of Structured Secondary Data for Health Research. Front Public Health. 2021 Jun 15;9:642163. doi: 10.3389/fpubh.2021.642163.

* RECORD 6.3: If the study involved linkage of databases, consider use of a flow diagram or other graphical display to demonstrate the data linkage process, including the number of individuals with linked data at each stage.

The deterministic data linkage procedures used in this study have been previously described in detail. Accordingly, we referenced the original methodological source (Reference #10), which provides a comprehensive description of the linkage process.

Waldvogel BC, Morais LCC, P

---

## [Editor Report · Decision Letter 1]

2 Feb 2026

A two-tiered latent class and spatial analytical approach to identify clusters of neonatal mortality among very low birth weight infants: a population-based cohort study

PONE-D-25-47659R1

Dear Dr. Costa-Nobre,

We’re pleased to inform you that your manuscript has been judged scientifically suitable for publication and will be formally accepted for publication once it meets all outstanding technical requirements.

Kind regards,

Ricardo Q. Gurgel, PhD

Academic Editor

PLOS One
---

## [Editor Report · Acceptance letter]

PONE-D-25-47659R1

PLOS One

Dear Dr. Costa-Nobre,

I'm pleased to inform you that your manuscript has been deemed suitable for publication in PLOS One. Congratulations! Your manuscript is now being handed over to our production team.

Kind regards,

on behalf of

Professor Ricardo Q. Gurgel

Academic Editor

PLOS One